# A Comparative Study about the Neuroprotective Effects of DHA-Enriched Phosphatidylserine and EPA-Enriched Phosphatidylserine against Oxidative Damage in Primary Hippocampal Neurons

**DOI:** 10.3390/md21070410

**Published:** 2023-07-19

**Authors:** Yi-Wen Wang, Qian Li, Xiao-Yue Li, Ying-Cai Zhao, Cheng-Cheng Wang, Chang-Hu Xue, Yu-Ming Wang, Tian-Tian Zhang

**Affiliations:** 1College of Food Science and Engineering, Ocean University of China, Qingdao 266404, China; wangyiwen@stu.ouc.edu.cn (Y.-W.W.); lq013493@163.com (Q.L.);; 2Laboratory for Marine Drugs and Bioproducts, Pilot National Laboratory for Marine Science and Technology (Qingdao), Qingdao 266237, China

**Keywords:** Alzheimer’s disease, oxidative stress, DHA/EPA, phosphatidylserine, primary hippocampal neurons

## Abstract

Nerve damage caused by accumulated oxidative stress is one of the characteristics and main mechanisms of Alzheimer’s disease (AD). Previous studies have shown that phosphatidylserine (PS) rich in eicosapentaenoic acid (EPA) and docosahexaenoic acid (DHA) plays a significant role in preventing and mitigating the progression of AD. However, whether DHA-PS and EPA-PS can directly protect primary hippocampal neurons against oxidative damage has not been studied. Here, the neuroprotective functions of DHA-PS and EPA-PS against H_2_O_2_/t-BHP-induced oxidative damage and the possible mechanisms were evaluated in primary hippocampal neurons. It was found that DHA-PS and EPA-PS could significantly improve cell morphology and promote the restoration of neural network structure. Further studies showed that both of them significantly alleviated oxidative stress-mediated mitochondrial dysfunction. EPA-PS significantly inhibited the phosphorylation of ERK, thus playing an anti-apoptotic role, and EPA-PS significantly increased the protein expressions of p-TrkB and p-CREB, thus playing a neuroprotective role. In addition, EPA-PS, rather than DHA-PS could enhance synaptic plasticity by increasing the expression of SYN, and both could significantly reduce the expression levels of p-GSK3β and p-Tau. These results provide a scientific basis for the use of DHA/EPA-enriched phospholipids in the treatment of neurodegenerative diseases, and also provide a reference for the development of related functional foods.

## 1. Introduction

Alzheimer’s disease (AD) is a progressive age-related neurodegenerative disease characterized by impaired cognitive function, memory loss, and behavioral and personality changes that have dramatic consequences for individuals and society [1,2]. Currently, there are about 50 million AD patients worldwide, and this number is expected to double every five years, rising to 152 million by 2050 [3]. Alzheimer’s has a long incubation period of 15 to 20 years, but current treatments only improve symptoms [4,5].

The histopathological hallmarks of AD are the extracellular formation of senile plaques composed of the β-amyloid (Aβ) peptide in an aggregated form along with metal ions such as copper, iron or zinc and the intracellular aggregation of neurofibrillary tangles [6,7]. Redox active metal ions, such as copper, can catalyze the production of reactive oxygen species (ROS) when bound to the Aβ. The produced ROS, in particular the hydroxyl radical, which is the most reactive one, may contribute to oxidative damage on both the Aβ peptide itself and on surrounding molecules, such as proteins and lipids [7]. Therefore, nerve damage caused by accumulated oxidative stress is one of the characteristics and main mechanisms of AD [8,9]. Therefore, improving oxidative stress is considered to be an effective therapeutic strategy to prevent and treat AD.

Among the nutrients that address the specific cognitive decline of aging, long-chain omega-3 polyunsaturated fatty acids (ω-3 LCPUFA), especially docosahexaenoic acid (DHA) and eicosapentaenoic acid (EPA), have emerged as very promising nutrients [10]. Studies have shown that EPA-enriched phosphatidylcholine (EPA-PC) rather than the ethyl ester (EE) form has a comparable effect with DHA-EE in improving cognitive impairment in Aβ1–42-induced AD rats [11]. Behavior test results indicated that DHA-enriched phosphatidylcholine (DHA-PC) exerted better effects than EPA-PC on improving memory and cognitive deficiency [12]. A growing body of research indicates that phosphatidylserine (PS) has significant effects in preventing and alleviating the progression of AD [13]. Our previous research found that both DHA-PC and DHA-enriched phosphatidylserine (DHA-PS) could ameliorate oxidative stress, and DHA-PS had more significant benefits in ameliorating Aβ pathology, mitochondrial damage, neuroinflammation, and improving the expression levels of neurotrophic factors than DHA-PC in a high-fat-diet-induced SAMP8 mouse model of Alzheimer’s disease [14]. The hippocampus, which is mainly responsible for learning and memory, is one of the most vulnerable parts of the brain and is particularly vulnerable to damage in the early stages of AD [8,15]. DHA-enriched phosphatidylserine (DHA-PS) and EPA-enriched phosphatidylserine (EPA-PS) could significantly eliminate toxicity in Aβ-induced primary hippocampal neurons by inhibiting mitochondria-dependent apoptotic pathways and the phosphorylation of JNK and p38, which significantly promoted axonal growth [16]. CHO-APP/PS1 cells have been widely used in AD models in vitro due to their ability to produce Aβ [12]. Phagocytosis by mouse small glioma cells (Bv-2 cells), as a kind of microglia, is proposed as an Aβ-lowering mechanism to prevent senile plaque accumulation in AD [17]. Moreover, the improved phagocytosis ability of microglia cells can effectively reduce Aβ deposition, which is beneficial for the treatment and prevention of AD [16]. Notably, DHA-PS could significantly reduce the production of Aβ in CHO-APP/PS1 cells compared with EPA-PS; on the contrary, EPA-PS significantly improved the phagocytic capacity of BV2 cells to Aβ compared with DHA-PS [16]. However, there is no study on whether DHA-PS and EPA-PS can directly protect primary hippocampal neurons against oxidative damage.

In this study, we evaluated the neuroprotective function of DHA-PS and EPA-PS against H_2_O_2_/t-BHP-induced oxidative damage and the possible mechanism in primary hippocampal neurons.

## 2. Results and Discussion

### 2.1. Morphology and Purity of Primary Cultured Hippocampal Neurons

The morphological changes of primary cultured neurons were investigated using an inverted microscope (Figure 1). The newly inoculated primary hippocampal neurons showed suspension, a round state and a small cell size. After 12 h, the adherent neuron cells were mostly fusiform, accompanied by the growth of protrusions in some cells. However, no obvious links between the cells were established. On the third day of culture, connections between cells began to form, along with the growth of neuronal cell processes and the enlargement of cell bodies. After six days, the cultured neurons were thriving, with more and longer protrusions extending into the reticulation. The purity of hippocampal neurons was about 95% using microtubule-associated protein 2 as a neuron-specific marker for the detection of primary cultured neurons [18].

### 2.2. The Effects of EPA-PS and DHA-PS on H_2_O_2_-Induced or t-BHP-Induced Morphological Damage in Primary Hippocampal Neurons

According to our previous publications, 10 μg/mL EPA-PS, 10 μg/mL DHA-PS, 400 μmol/L H_2_O_2_ and 100 μmol/L t-BHP were selected as the concentrations for subsequent experiments [8,16]. In this study, we used a microscope to study the protective function of DHA-PS and EPA-PS on the H_2_O_2_- or t-BHP-induced oxidative damage of hippocampal neurons. As shown in Figure 2, in the control group, the neuron body was full, with more and longer protrusions, and the cell refraction was good, while the boundary was neat and smooth. When treated with H_2_O_2_ or t-BHP, the neurites of neurons became significantly shorter, with the swelling of cells, atrophy of the cytoplasm, a reduced refractive index and damage to parts of the neuronal membrane. Compared with the model group, after incubation with EPA-PS and DHA-PS, the cell membrane boundary became neat and smooth, synapses were restored, and the neural network structure was improved. Phospholipids are the most important lipid components in the brain and have neuroprotective effects [19]. The sn-2 site of marine-derived phospholipids is rich in polyunsaturated fatty acids, making it more biologically active in brain function [13]. PS is a major anionic phospholipid substance, enriched in the cerebral cortex [20], which is involved in the regulation of neurotransmitter release and neuron survival and differentiation [20,21]. It has been reported that the EPA-PS and DHA-PS treatments significantly decreased neuronal death induced by Aβ42, and the improvement effect of DHA-PS is more pronounced [16]. In addition, the results of our study may be affected by impurities because fewer than 5% of the impurities are contained in DHA-PS and EPA-PS [16].

### 2.3. Effects of EPA-PS and DHA-PS on the Expression of Proteins Related to Apoptosis in Primary Hippocampal Neurons

Mitochondria-dependent apoptosis pathways mainly include upstream BAX/Bcl-2, which is the switch of the mitochondria-dependent apoptosis pathway, and downstream caspase family [22]. BAX has the function of promoting cell apoptosis, while Bcl-2 acts as an inhibitor of the apoptotic pathway [16]. The BAX/Bcl-2 ratio is an important indicator of the activation of mitochondria-dependent apoptosis [23]. Caspase-9, an initiator, can be activated by Cyt-c released from the mitochondria. Moreover, caspase-3 is an executioner and distinctly predominates in neurodegenerative diseases [16]. Studies have shown that this pathway is significantly activated after the nervous system is damaged by oxidative stress [24].

The effects of EPA-PS and DHA-PS on the mitochondrial-dependent apoptotic pathways after oxidative damage are shown in Figure 3. Compared with the control group, the expression level of BAX protein was significantly increased after oxidative damage. The proapoptotic BAX protein expression level was decreased by 47.8% (*p* < 0.01) and 44.9% (*p* < 0.01) in the EPA-PS group and DHA-PS group, respectively. Compared with the control group, the expression level of Bcl-2 protein was not significantly affected after oxidative damage, but the expression level of Bcl-2 was increased by 91.8% (*p* < 0.01) and 21.4% (*p* < 0.05) in the EPA-PS group and DHA-PS group, respectively. Oxidative damage significantly upregulated the BAX/Bcl-2 ratio—that is, oxidative damage significantly activated mitochondria-dependent apoptotic signaling switches, and both EPA-PS and DHA-PS interventions significantly reduced BAX/Bcl-2 levels, among which the effect of EPA-PS was better than that of DHA-PS. Compared with the control group, the protein expression levels of Caspase 9 and Caspase 3 in the model group were not significantly changed, but the protein levels of Caspase 9 and Caspase 3 were significantly reduced after incubation with EPA-PS or DHA-PS. In addition, DHA-PS was superior to EPA-PS in inhibiting Caspase 3 expression. Treatments with DHA-PS and EPA-PS markedly inhibited oxidative stress-mediated mitochondrial dysfunction. These data indicated that DHA-PS and EPA-PS exerted their neuroprotective properties by inhibiting the neuronal apoptosis. Interestingly, Xu et al. [16] reported that both DHA-PS and EPA-PS significantly reduced the BAX/Bcl-2 expression ratio of Aβ-induced primary hippocampal neurons, but had no significant effect on the protein expression of caspase 3 and caspase 9. Additionally, according to Che et al. [18], compared with EPA-PS, DHA-PS had a better protective effect against the oxidative stress-induced cell damage of pheochromocytoma cells, and could more significantly reduce the expression ratio of BAX/Bcl-2, but both reduced the mRNA abundance of caspase 3 and caspase 9 to the same degree. The differences in the regulation of DHA-PS and EPA-PS on the apoptosis proteins may be related to the cell model and incubation dose.

### 2.4. Effects of EPA-PS and DHA-PS on the TrkB/ERK/CREB Signaling Pathway and Synaptic Associated Proteins

It has been shown that cyclic adenosine monophosphate-dependent response element-binding protein (CREB) plays a key role in neuronal plasticity and is mainly regulated by brain-derived neurotrophic factor (BDNF). The binding of BDNF to TrkB results in the self-phosphorylation of TrkB and the activation of its downstream enzymes, including extracellular signal-regulated kinase (ERK) and PI3K/Akt pathways [25,26]. At the same time, ERK is an extracellular regulatory protein kinase. After the extracellular stimulus acts on cells, the corresponding biological effects must be triggered through the ERK signal transduction pathway. Therefore, the level of p-ERK increases significantly when exogenous injurers are applied to cells, and phosphorylated ERK may mediate neuronal cell apoptosis by activating p-53 [18]. Phosphorylated CREBs improve synaptic plasticity by upregulating synapse-related proteins such as PSD-95 and SYN [27].

Compared with the control group, the protein expression levels of p-CREB and p-TrkB did not change significantly after oxidative damage (Figure 4). Interestingly, the protein expression level of p-TrkB was increased by 105% (*p* < 0.01) and 40% (*p* < 0.05) after incubation with EPA-PS and DHA-PS, respectively. Compared with the model group, EPA-PS incubation increased p-CREB protein expression to 158.9% (*p* < 0.05), while in the DHA-PS group, p-CREB protein expression was not significantly affected. Compared with the control group, the protein expression level of p-ERK in the model group was significantly increased (*p* < 0.05), and the protein expression level of p-ERK was significantly decreased after incubation with EPA-PS and DHA-PS, and the effect of EPA-PS was better than that of DHA-PS (Figure 4). It is suggested that EPA-PS and DHA-PS play an anti-apoptotic role by inhibiting ERK phosphorylation. Interestingly, DHA-PS and EPA-PS had no significant effect on the expression of TrkB and CREB in primary hippocampal neurons induced by Aβ [16]. The different results might be associated with the different causes of neuronal damage, which need further study.

As shown in Figure 5, SYN protein expression in the model group was not changed significantly compared with the control group. SYN expression was increased to 130.3% after incubation with EPA-PS (*p* < 0.05), but was not significantly affected after incubation with DHA-PS. There was no significant difference in the protein expression of PSD-95 among all groups. These results indicate that EPA-PS could improve synaptic plasticity after oxidative damage. GAP-43 regulates the growth state of axon terminals, and the deficient expression of GAP-43 results in the weak neuron regeneration ability after damage, which is linked to long-term potentiation [14]. Similarly, it was reported that EPA-PS significantly increased the expression of both GAP-43 (2.05-fold) and SYN (1.83-fold) compared with the SAMP8 group [28]. Interestingly, in another study, Zhou et al. also found that DHA-PS significantly increased the expression levels of growth-related protein-43 (GAP-43) and SYN in SAMP8 mice, and improved the cognitive impairment of SAMP8 mice [14]. Surprisingly, EPA-PS and DHA-PS had no significant effect on the expression of PSD-95 and SYN in Aβ-induced primary hippocampal neurons [16]. Therefore, it is speculated that the improving effects of DHA-PS and EPA-PS on the neurites of neurons may be influenced by inducible factors and experimental animals/cells.

### 2.5. Effects of EPA-PS and DHA-PS on Tau Protein Phosphorylation

Tau is a microtubule-binding protein that stabilizes tubulin to form microtubules under physiological condition [29]. However, in the brains of AD patients, the activation of GSK3β kinase results in the excessive phosphorylation of Tau proteins, which leads to the loss of normal physiological function, resulting in axon transport dysfunction, the loss of synapses, and ultimately neuronal death [30,31]. Therefore, the levels of Tau phosphorylation and its major kinase GSK3β were measured (Figure 6). The results of this study show that the phosphorylation level of Tau protein increased significantly after oxidative damage, which was consistent with the results of previous studies [32]. Compared with the model group, EPA-PS and DHA-PS significantly reduced the protein expression levels of p-GSK3β and p-Tau to the same extent. These results suggest that there was no difference between DHA-PS and EPA-PS in improving nerve fiber tangles after oxidative damage. Similarly, studies have shown that EPA-PS can inhibit the level of p-GSK3β in the brain hippocampus of aged SAMP8 mice, and also significantly reverse the phosphorylation of Tau 0.62-fold [28].

### 2.6. Effects of EPA-PS and DHA-PS on the PI3K/Akt Signaling Pathway

The PI3K/Akt pathway is widely found in cells and is a signal transduction pathway involved in cell growth, proliferation and differentiation, playing an important role in promoting cell survival during oxidative stress [33,34]. PI3K/Akt kinase cascade is an important pathway for membrane receptor signal transduction into cells [35]. PI3K is an intracellular phosphatidyl inositol kinase. The activation of PI3K phosphorylates Akt at Ser 473, thereby activating Akt, which plays a role in promoting cell survival by regulating apoptosis-related proteins [34]. As shown in Figure 7, the protein expression level of p-PI3K was significantly increased after oxidative damage. After incubation, the protein expression level of p-PI3K was not changed significantly in the EPA-PS group, while it was significantly decreased by 18.5% (*p* < 0.05) in the DHA-PS group. Compared with the control group, p-Akt protein expression levels in the model group were not changed significantly. Compared with the model group, the expression level of p-Akt protein was significantly up-regulated after the incubation of EPA-PS, while that of p-Akt protein was not significantly affected in the DHA-PS group. Interestingly, in primary hippocampal neurons induced by Aβ, p-PI3K levels were decreased and p-Akt levels were increased. Compared with the Aβ-induced group, p-PI3K levels in EPA-PS and DHA-PS groups did not change significantly, but EPA-PS and DHA-PS significantly reduced p-Akt levels by 16.8% and 23.8%, respectively [16]. In this study, we found that the levels of p-Akt and p-PI3K protein increased significantly after oxidative stress injury, which might be related to the stress response of cells to external stimuli. Accordingly, our current research suggested that EPA-PS and DHA-PS administration might protect primary hippocampal neurons after oxidative damage, but not through the PI3K/Akt pathway.

PS influx into absorptive cells occurs after its hydrolysis to lysoPS, and lysoPS, after diffusion into intestinal cells, is sequentially converted into PS and PE, which make up a minor fraction of the lipids present in lipoproteins [36]. The brain is one of the tissues with high capacity to synthesize PS [37]. In mammalian tissues, PS is synthesized from either PC or PE exclusively by Ca^2+^-dependent reactions, where the head group of the substrate phospholipids is replaced by serine [38]. These base-exchange reactions are catalyzed by phosphatidylserine synthases (PSS) [20], which are localized in the endoplasmic reticulum, particularly enriched in the mitochondria-associated membrane regions of the endoplasmic reticulum [39]. Therefore, dietary DHA-PS and EPA-PS do not enter primary hippocampal neurons in their original form, but are synthesized again after digestion and absorption. It has been reported that oral PS was absorbed efficiently in humans and crossed the blood–brain barrier following its absorption into the bloodstream, increasing the content of PS in the brain [40,41] and its incorporation into neuron cell membranes [42]. The incorporation of adequate amounts of PS within nerve cell membranes is required for efficient neurotransmission throughout the human nervous system [42]. In addition, our previous study found that a high-fat diet significantly reduced the levels of PS/pPE containing DHA, PS containing DPA, and PE containing AA in the cerebral cortex of SAMP 8 mice, but dietary DHA-PC and DHA-PS significantly restored lipid homeostasis [43]. Therefore, although DHA-PS and EPA-PS do not enter the hippocampus in their original form, the dietary supplementation of DHA-PS and EPA-PS is still significant for hippocampal neurons.

The neuroprotective effects of DHA-PS and EPA-PS against oxidative damage in primary hippocampal neurons is shown in Figure 8. There are some other limitations to the study. PS is normally found on the inner leaflet of the plasma membrane, and the ATP-driven aminophospholipid translocases ATP8A1 and ATP8A2 pump phosphatidylserine from the outer side of the membrane to the inner side of the membrane [44]. Interestingly, PS can be exposed on the cell surface and is associated with apoptosis-related proteins and receptor proteins that cause microglia to undergo phagocytosis [45]. Moreover, the anti-inflammatory microglial phenotype induced through the activation of the specific PS receptor (PtdSerR), expressed by resting and activated microglial cells, could be relevant to the final outcome of neurodegenerative diseases, in which apoptosis seems to play a crucial role [46]. Based on the reports on PS receptors, further studies on DHA-PS and EPA-PS receptors will be conducted in the future. In addition, it was not determined whether DHA-PS and EPA-PS liposomes were fully incorporated into hippocampal cells after 24 h of preincubation, which may lead to the remaining liposomes that can be oxidized thus playing a role in reducing oxidative damage to cells. The cellular protection may be aided by the presence of extracellular EPA-PS and DHA-PS, a situation that will likely not occur in vivo in the brain, where lipids are transported and remodeled by physiological mechanisms and phospholipids cannot just be added as liposomes. Meanwhile, the oxidation status of the DHA-PS and EPA-PS liposomes was not determined, and analysis of this aspect will be carried out in follow-up studies.

## 3. Materials and Methods

### 3.1. Materials

Sea cucumber (*Cucumaria frondosa*) and squid (*Sthenoteuthis oualaniensis*) were obtained from Nanshan aquatic market of Qingdao (Qingdao, China). Dulbecco’s modified Eagle’s medium (DMEM) and fetal bovine serum (FBS) were purchased from GIBCO (Grand Island, NY, USA). 3-(4,5-Dimethylthiazol-2-yl)-2,5-diphenyltetrazolium bromide (MTT) was provided by Sigma-Aldrich (St Louis, MO, USA). RIPA Lysis Buffer, a BCA Protein Assay Kit, and phenylmethylsulfonyl fluoride (PMSF) were provided by Beyotime (Shanghai, China). Antibodies of BAX, Blc-2, Caspase 9, Caspase 3, Cyt-c, phosphorylated TrkB (Y515), phosphorylated ERK and phosphorylated CREB, synaptophysin (SYN), postsynaptic compacts (PSD-95), phosphorylated GSK3β, phosphorylated Tau, phosphorylated phosphatidylinositol 3 kinase (PI3K) and phosphorylated Akt were obtained from Cell Signaling Technology (Boston, MA, USA).

### 3.2. Preparation of EPA-PS and DHA-PS

EPA-enriched phosphatidylcholine (EPA-PC) and DHA-PC were extracted from sea cucumber (*C. frondosa*) and squid roe, respectively, based on the modified method of Folch [47,48]. Briefly, the freeze-dried samples were immersed in a 20-fold volume of chloroform-methanol solution (2:1, *v*/*v*) for 12 h. Then, water was added to the filtered extraction solution. The mixture was poured into a separatory funnel and the underlying solution containing chloroform and total lipid was collected. Next, the collected solution was evaporated under vacuum until dry to remove the organic solvent. Then, EPA-PC and DHA-PC were separated from neutral lipids and glycolipids by means of silica-gel column chromatography using chloroform, acetone, chloroform-methanol (9:1, *v*/*v*), chloroform-methanol (2:1, *v*/*v*) and methanol sequentially as eluents. The methanol eluent was collected, and EPA-PC and DHA-PC were obtained after removal of the organic solvent under vacuum. PS was synthesized from the PC via the phospholipase D catalyzed transphosphatidyl reaction based on the previously described method [49]. The purities of EPA-PS and DHA-PS were found to be above 90% using high-performance liquid chromatography [16].

### 3.3. Preparation of EPA-PS and DHA-PS Liposomes

Liposomes were prepared according to previously reported methods with slight modifications [50]. Concisely, a mixture of phosphatidylserine (2 mg) and cholesterol (molar ratio 1:1) was dissolved in chloroform, and then dried under a nitrogen stream to form thin films in a rotary flask. A certain amount of phosphate-buffered solution was added to the flask and nitrogen protection was performed. It was dissolved using a vortex at low temperature and became a uniform suspension with a concentration of 1 mg/mL (calculated by phospholipids). The obtained liposome suspension was extruded 20 times through 400 nm and 200 nm polycarbonate membrane filters and then stored at −20 °C. EPA-PS and DHA-PS liposomes were prepared fresh every time.

### 3.4. Preparation of Primary Hippocampal Neurons

P0 pups of Sprague Dawley (SD) rats were provided by Qingdao Lukang Pharmaceutical Experimental Animal Center. According to the method previously reported, hippocampal neurons were isolated from the brains of neonatal SD rats [51]. The hippocampal tissues were divided into 1 mm^3^ pieces and digested with 0.125% trypsin in a CO_2_ incubator for 15 min. DMEM medium containing 10% FBS was added to the hippocampal tissue, and the upper cell fluid was collected after standing. B-27 (2%) was added on the third day of culture and the medium was replaced every other day thereafter [16].

### 3.5. Morphological Observation of Primary Hippocampal Neurons

The prepared liposomes with a concentration of 1 mg/mL were diluted to 10 μg/mL using the medium. The extracted hippocampal neurons were prepared in cell suspensions and inoculated into 24-well plates, and incubated with EPA-PS liposomes or DHA-PS liposomes at 10 μg/mL, respectively [16]. The morphology of hippocampal neurons was observed using an inverted microscope.

### 3.6. Western Blotting Analysis

Primary hippocampal neurons were lysed in a RIPA lysis buffer containing 1% PMSF. After incubation on ice for 30 min, the lysate was centrifuged at 8000 rpm for 10 min at 4 °C. Then, the amount of protein in the solution was measured using a BCA protein assay kit (Beyotime, Nanjing, China). Cellular proteins (20 µg) were isolated using 10% sodium dodecyl sulfate-polyacrylamide gelelectrophoresis. Then, proteins were transferred onto polyvinylidene fluoride membranes, which were blocked in 5% (*w*/*v*) bovine serum albumin (BSA) for 2 h at room temperature and then incubated with primary antibodies (BAX (1:2000), Blc-2 (1:2000), Caspase 9 (1:1000), Caspase 3 (1:1000), Cyt-c (1:1000), p-TrkB (1:1000), p-ERK (1:1000), p-CREB (1:1000), SYN (1:1000), PSD-95 (1:2000), p-GSK3β (1:1000), p-Tau (1:1000), p-PI3K (1:1000) and p-Akt (1:1000)) overnight at 4 °C. Next, the membranes were incubated with a goat anti-rabbit immunoglobulin G secondary antibody for 2 h at room temperature. Finally, enhanced chemiluminescence was used to visualize the corresponding protein bands with an UVP Auto Chemi Image system. Protein loading was evaluated by using anti-β-actin antibody (1:3000).

### 3.7. Statistical Analysis

Data are expressed as means ± SEM, and the reported values are representative of three independent experiments. The comparison between the control and model groups was carried out using Student’s test * *p* < 0.05. Differences among the three groups, model, EPA-PS and DHA-PS, were tested via one-way ANOVA (Turkey’s test), and different letters (a, b, c) indicated different significance at *p* < 0.05.

## 4. Conclusions

In this study, we found that DHA-PS and EPA-PS could significantly improve abnormal cell morphology and promote the restoration of the neural network structure. Both of them could significantly reduce and inhibit oxidative stress-mediated mitochondrial dysfunction, but the inhibition effect of DHA-PS on the expression of Caspase 3 was better than that of EPA-PS, while the reduction effect on BAX/Bcl-2 level was the opposite. Both of them could play a neuroprotective role by regulating the TrkB/ERK/CREB pathway, and the protective effect of EPA-PS was more obvious. In addition, EPA-PS could improve the synaptic plasticity after oxidative damage by increasing the expression of SYN, but DHA-PS had no effect on the expression of SYN and PSD-95. In addition, both could improve ganglion entanglement by significantly reducing the expression levels of p-GSK3β and p-Tau to the same extent.

## Figures and Tables

**Figure 1 marinedrugs-21-00410-f001:**
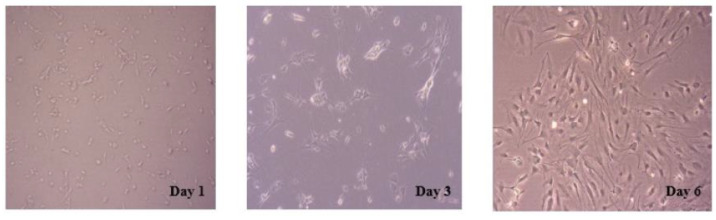
Morphology of primary cultured hippocampal neurons at different stages. The representative images of primary hippocampal neurons on day 1, day 3 and day 6 were observed using an inverted microscope.

**Figure 2 marinedrugs-21-00410-f002:**
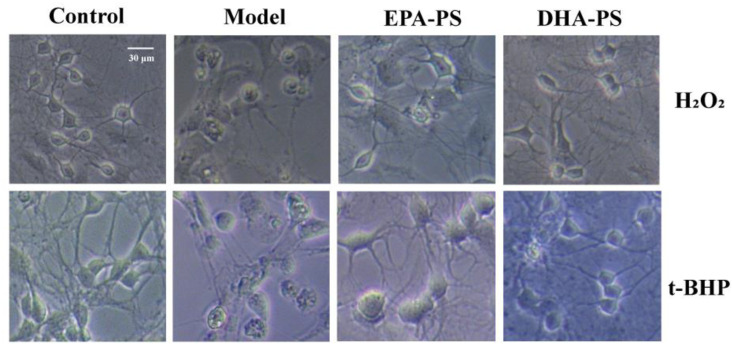
Effects of EPA-enriched phospholipids on the cellular morphology of primary hippocampal neurons treated with EPA-PS or DHA-PS for 24 h then exposed to 400 μmol/L H_2_O_2_ for 24 h or 100 μmol/L t-BHP for 4 h. Cellular morphology was observed using a microscope.

**Figure 3 marinedrugs-21-00410-f003:**
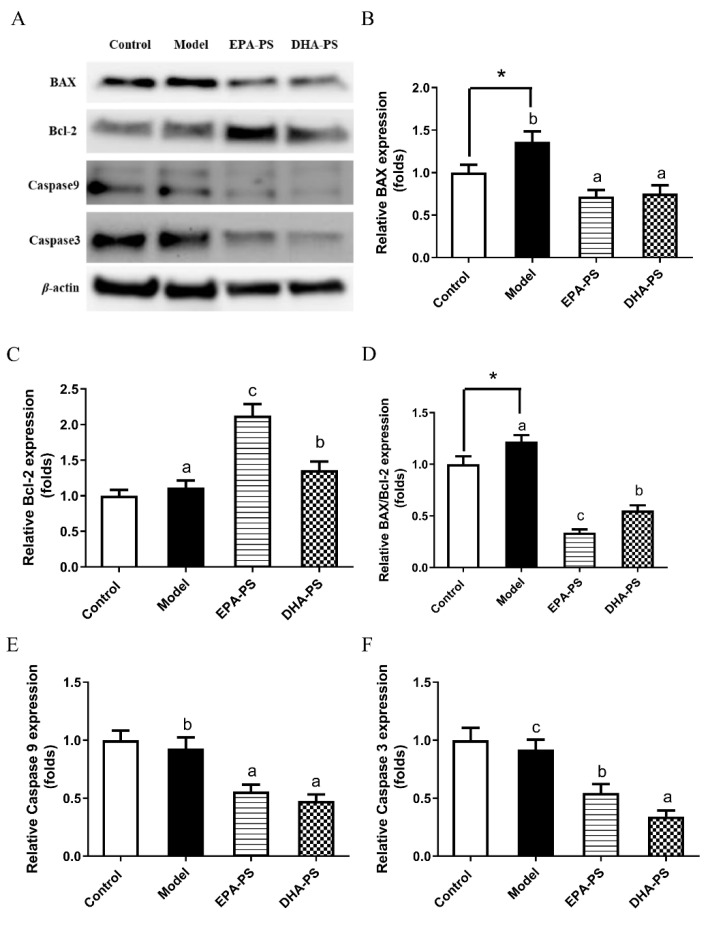
Effects of EPA-PS and DHA-PS on the expression of apoptotic proteins in primary hippocampal neurons after oxidative damage. Representative western blots (**A**) and relative expression of BAX (**B**), Blc-2 (**C**), BAX/Blc-2 (**D**), Caspase 9 (**E**) and Caspase 3 (**F**) in primary hippocampal neurons. The expressions were detected by Western blotting analysis and normalized with β-actin. * *p* < 0.05 indicates significant differences compared with the control group. Different letters represent significant differences at *p* < 0.05 among treated groups. This figure shows the mean ± SEM of 3 experiments.

**Figure 4 marinedrugs-21-00410-f004:**
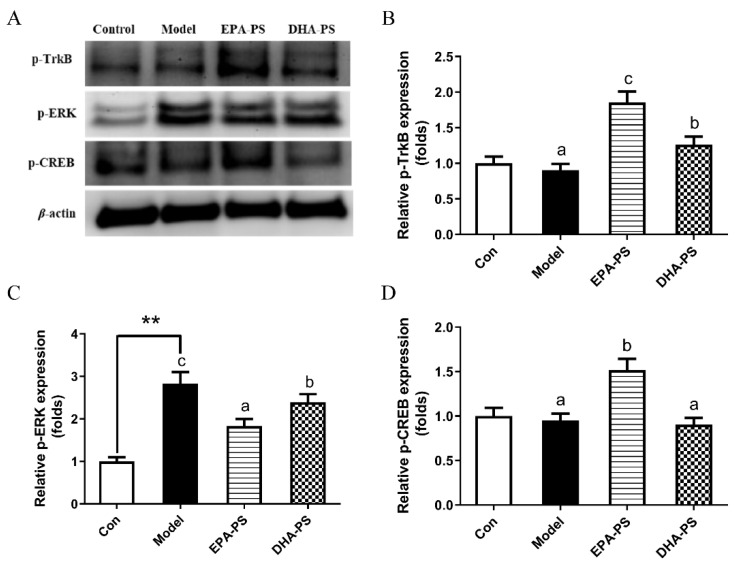
Effects of EPA-PS and DHA-PS on TrkB/ERK/CREB signaling in neurons after oxidative damage. Representative western blots (**A**) and relative expression of phosphorylated TrkB (Y515) (**B**), phosphorylated ERK (**C**) and phosphorylated CREB (**D**) in primary hippocampal neurons. The expressions were detected by Western blotting analysis and normalized with β-actin. ** *p* < 0.01 indicates very significant differences compared with the control group. Different letters represent significant differences at *p* < 0.05 among treated groups. This figure shows the mean ± SEM of 3 experiments.

**Figure 5 marinedrugs-21-00410-f005:**
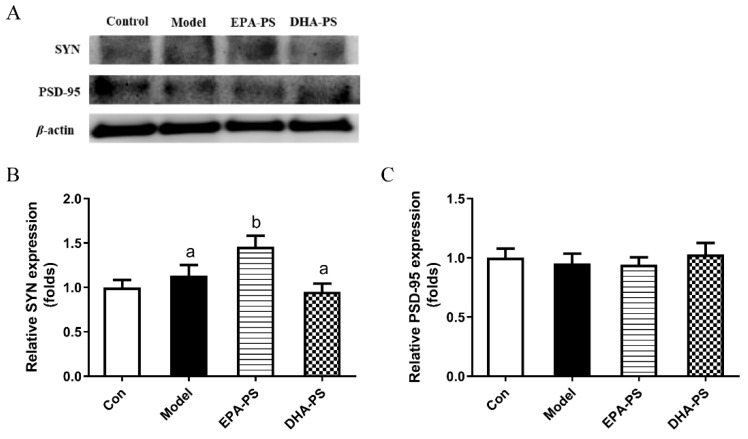
Effects of EPA-PS and DHA-PS on the expressions of SYN and PSD-95 in neurons after oxidative damage. Representative western blots (**A**) and relative expression of SYN (**B**) and PSD-95 (**C**) in primary hippocampal neurons. The expressions were detected by Western blotting analysis and normalized with β-actin. Different letters represent significant differences at *p* < 0.05 among treated groups. This figure shows the mean ± SEM of 3 experiments.

**Figure 6 marinedrugs-21-00410-f006:**
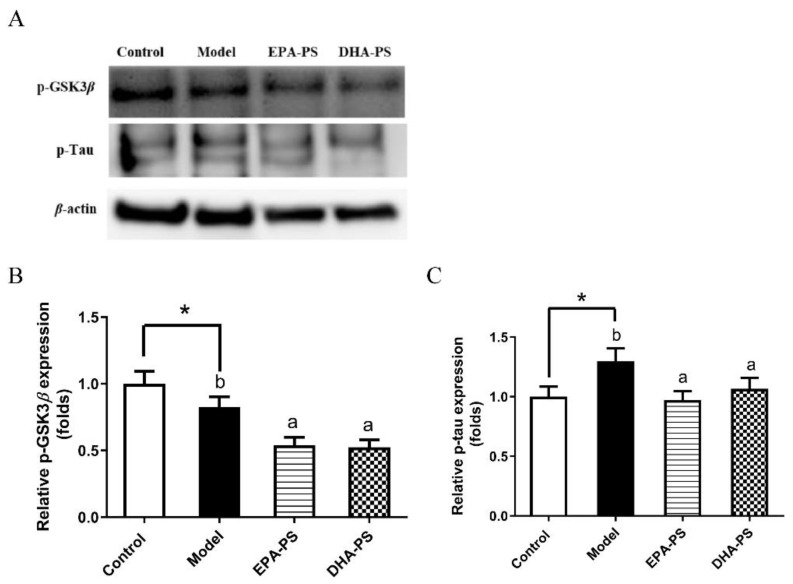
Effects of EPA-PS and DHA-PS on the expressions of GSK3β and Tau in neurons after oxidative damage. Representative western blots (**A**) and relative expression of p-GSK3β (**B**) and p-Tau (**C**) in primary hippocampal neurons. The expressions were detected by Western blotting analysis and normalized with β-actin. * *p* < 0.05 indicates significant differences compared with the control group. Different letters represent significant differences at *p* < 0.05 among treated groups. This figure shows the mean ± SEM of 3 experiments.

**Figure 7 marinedrugs-21-00410-f007:**
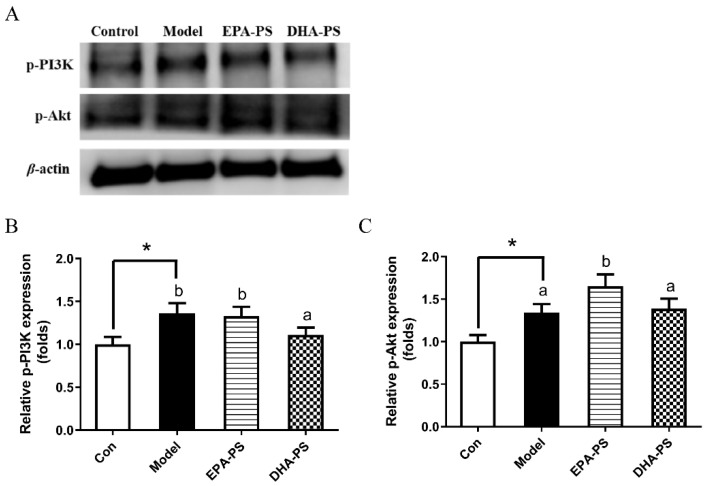
The effects of EPA-PS and DHA-PS on the PI3K/Akt signaling pathway in neurons after oxidative damage. Representative western blots (**A**) and relative expression of p-PI3K (**B**) and p-AKT (**C**) in primary hippocampal neurons. The expressions were detected by Western blotting analysis and normalized with β-actin. * *p* < 0.05 indicates significant differences compared with the control group. Different letters represent significant differences at *p* < 0.05 among treated groups. This figure shows the mean ± SEM of 3 experiments.

**Figure 8 marinedrugs-21-00410-f008:**
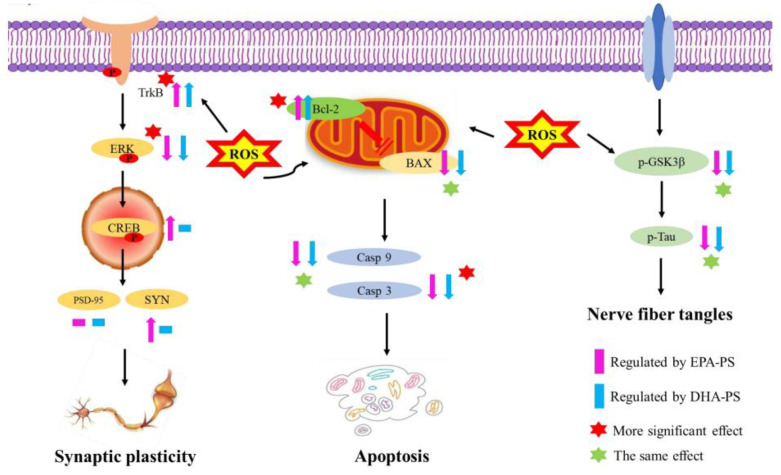
Possible potential mechanisms for the neuroprotective effects of EPA-PS and DHA-PS on the oxidative damage of primary hippocampal neurons.

## Data Availability

Not applicable.

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
