# Peer review of "A Comparative Study about the Neuroprotective Effects of DHA-Enriched Phosphatidylserine and EPA-Enriched Phosphatidylserine against Oxidative Damage in Primary Hippocampal Neurons"

_marinedrugs, 2023, doi:10.3390/md21070410_

Round 1
Reviewer 1 Report
The study investigated the potentials of EPA-PS and DHA-PS in the treatment of neurodegenerative disease (Alzheimer’s Disease). The manuscript needs major revision.
1) The work did not study how and what receptors of EPA-PS or DHA-PS mediated the regulation of cell functions.
2) The authors did not compare with DHA or EPA. It is not clear why EPA-PS or DHA-PS was better and EPA or DHA or why PS is important in regulating cell apoptosis.
3) The authors need to do the statical analysis in all figures. In addition, the figure captions are not clear to describe how to get the results and what experimental conditions.
4) It is confusing that the authors made liposomes of EPA-PS or DHA-PS in Materials and Experiments, but it is not clear whether the authors used liposomes or EPA-PS or DHA-PS in the experiments.
5) There is no animal study in the manuscript, but the authors included the animal study section. It is needed to perform animal studies to prove the benefit of EPA-PS or DHA-PS.
6) Around Line 140-149, the explanation given on the impact of DHA-PS and EPA-PS on the expression of caspase 8 and caspase 3 seems contradictory. This is because the first sentence mentioned that the protein levels of the two caspases were reduced but the following sentence noted that there is no significant change at the protein level. Although the mRNA only impact was later mentioned, mitigating confusion for the reader would be avoided if this is stated clearly where necessary.
no comment
Reviewer 2 Report
I have reviewed the manuscript by Wang and colleagues. This study has addressed specific biochemical changes in response to exposure of primary rat hippocampal neurons with EPA-phosphatidyl serine (PS) and DHA-PS that are stressed by the oxidants hydrogen peroxide and the organic peroxide tert-butyl-hydroperoxide. The results are clearly presented, and provide new insight for this cell type. I have a number of major and minor comments that the authors can address to improve their manuscript.
Major comments
1. From several statements in the manuscript the authors make us believe that phosphatidyl-serine containing the omega-3 long chain polyunsaturated fatty acids (LCPUFA) EPA and DHA are transported intact all the way from oral intake to the membranes of neuronal cells in the hippocampus. For example in line 39 there is reference to “functional food ingredients”, line 78 mentions that the results provide a theoretical basis for the application of DHA- and EPA-PS in the prevention and treatment of neurodegenerative disease, and line 350 mentions that the findings from this study may even provide dietary guidance for the prevention of Alzheimer’s Disease and a reference for the development of functional foods. Such statements find no support in this study, and I think this language needs to be removed from the manuscript and the authors just focus on reporting the findings of their study. Or if you want to discuss these claims then provide evidence for that in the Discussion section:
Please provide literature references that phosphatidyl-serine is not hydrolysed after oral intake in the digestive tract, is absorbed intact by enterocytes, is transported across the blood brain barrier, and that it is the original EPA-PS and DHA-PS that ends up in the hippocampus. If that is not possible, and I think there is little proof for that scenario today, you cannot make conclusions or suggestions that these phospholipids are the same as what is found in food or supplementation, and are not formed during local remodelling or biosynthesis from precursors in the brain itself.
Also in line 107 you refer to the predominance of LCPUFA in the sn2 position of marine-derived phospholipids and that that positional specificity is the reason that these phospholipids are more biologically active in the brain. But is the whole phospholipid absorbed and transported intact to the brain? In fact, in recent years, it has been described that it is the lyso-phospholipid form of PC that is a main form of transporting DHA and EPA into the brain via the MFSD2A transporter, and not the intact diacylated phospholipids. Please expand on this topic in your manuscript and limit yourself to discussing what you have observed in your experimental model.
2. Line 289 – EPA-PS and DHA-PS were prepared from biological sources in combination with enzymatic synthesis. The fact that the test compounds were only 90% pure should be recognized in the Discussion section, because it leaves open the possibility that the interesting biochemical events that are found in this study may be due to some other compounds that are a portion of the other 10%. Please also show a HPLC chromatogram of your EPA-PS and DHA-PS preparation, in order to see if there is a limited number or many other peaks present that constitute this other 10%.
3. Paragraph 3.3 – Apparently the EPA-PS and DHA-PS was delivered to the cells as liposomes. That should be stated in paragraph 3.6. In 3.3 it should also be explained how the concentration of EPA-PS and DHA-PS is quantified, in order to calculate that a 10 ug/ml final concentration is accurately delivered to the cells. Is it based on the weight of the entire molecules by mass spectrometric analysis of EPA-PS and DHA-PS, by phosphate determination, or based on the content of the EPA and DHA?
4. Paragraph 3.3 – Sonication of lipid films in PBS creates a great risk for oxidation of the esterified EPA and DHA. Did you measure the Peroxide Value of your final liposome preparations? How did you rule out that what you delivered to the cells is not a mixture of EPA-PS and DHA-PS and a large range of lipid peroxidation products, some of which may have potent biological activity. Also have you measured Free Fatty Acids or Acid Value, to determine how much hydrolysis has taken place (also free fatty acids have biological activity through FFA receptors).
5. Paragraph 3.5 – MTT reduction was employed as a read out of cell viability. However, MTT reduction is primarily a measure of mitochondrial activity and not cell death. First of all, I don’t see the results of cell viability measured by MTT reduction in the manuscript. Please include. And explain if you have measured cell death by other parameters as decreases in MTT reduction may precede cell death. This is important with respect to how you describe the effects of the two peroxides in your manuscript, often referred to as “oxidative damage” and not cell death.
6. Line 335 – please state how many experimental repetitions of the experiments contribute to the mean value and the sem. Also indicate the number of repetitions in the legends of each figure. “This figure shows the mean +- sem of n experiments, with n being the number.
7. A main aspect of this study that should be discussed and mentioned as an experimental limitation of the study by the authors is that when phospholipids are applied to the extracellular space of cells in culture, that there is no guarantee that these lipids are actually incorporated into the membranes of the cells. Unless you have measured and confirmed membrane incorporation, it is plausible that the EPA-PS and DHA-PS-rich liposomes are acting as a free radical trap for any peroxide-derived free radicals generated from the lipid peroxidation process initiated by exposure to the very high concentrations of hydrogen peroxide or tert-butyl hydroperoxide. There may also be other oxidizable lipids in the cell culture medium you use. In that case, the antioxidant effect by the extracellular liposomes redirecting free radicals away from the peroxidation that would happen in the cells is helping in reducing the oxidative damage. This scenario does not take away from the biochemical effects that you have found, but at least you should show that liposomal EPA-PS and DHA-PS is incorporated in the hippocampal cells during the 24 hours of preincubation. If you have not measured that, you need to discuss the limitations of your experimental study that the cellular protection may be aided by the presence of extracellular EPA-PS and DHA-PS, a situation that will likely not occur in vivo in the brain where lipids are transported and remodelled by physiological mechanisms and phospholipids can not just be added as liposomes.
In summary, I think your manuscript should much better explain the limitations of the interpretation of your findings, and not make wide extrapolations.
Minor comments
1. Line 68 – phagocytosis “by”
2. Please review the English language of line 71-74. “DHA-PS could significantly reduce the production of x than EPA-PS” is not correct English
3. It is not clear what is meant by “showed a small volume of suspension”. The volume of suspended cells is determined by the operator, no? Please rephrase
3. Line 89 – it is not clear what you mean by “the nervous process erupted”. Cell eruption is not a commonly used scientific terminology
4. Line 100 – There is no “normal” group in Figure 2. The legend in Fig 2 refers to “Control” or “Model”. Please better describe what you are referring to
5. In the top legend of Figure 2, it is not clear what you mean with “Model”. You mean the exposure to either of the two peroxides?
A few minor English language errors are noted in the comments to the authors above.
Round 2
Reviewer 2 Report
The manuscript has been improved. I have a few remaining issues that need to be addressed by the authors:
1. Line 297 – Please refer to the "inner leaflet" of the membrane, not "inner lobules"
2. Line 103-104 - It is not acceptable to say that the relevance of 5% impurities can be ignored. Scientists aim to work with the "pure substance", and if not available must recognize the possibility that the effects might be due to impurities, contaminants, metabolites and degradation products. Some oxidized lipids, free fatty acids, and oxylipins, for example can have extremely potent biological activity, with receptor agonist affinities in the picomolar and nanomolar range. Their activities cannot be ignored and could well be present in the 5% "impurity". The absence of pure EPA-PS and DHA-PS should be indicated as a limitation in the Discussion section.
3. Lines 352-353. It is still not clear how the amount or concentration (not the mass or molecular weight) of EPA-PS and DHA-PS is determined. Apart from knowing the molecular weight, you need to determine how much material you actually have available in your liposomes, in order to know how much to add to the cells. Please state the method you used to quantify these lipids.
4. Paragraph 3.3. - Highly unsaturated fatty acids such as EPA and DHA can start oxidizing within minutes when handled, especially in a phosphate buffer rich in transition metals, and using a vortex that may facilitate interfacial oxidation reactions. Although the authors say they worked under nitrogen, nitrogen blanking does not displace dissolved oxygen. It is essential to measure the oxidative quality of the final liposomal preparation to know to what extent the EPA and DHA have gotten oxidized, or to confirm they are not partially oxidized. Merely assuming that a fresh preparation does not lead to oxidation is naïve, these are extremely sensitive materials. You actually need to measure this. Please add a statement to the Discussion section, that the oxidative status of the EPA- and DHA rich liposomes was not determined, and this should be carried out in follow-up studies.
